# A Lightweight and Low-Voltage-Operating Linear Actuator Based on the Electroactive Polymer Polypyrrole

**DOI:** 10.3390/polym15163455

**Published:** 2023-08-18

**Authors:** Yeji Kim, Yasukazu Yoshida

**Affiliations:** Advanced Technology Research Department, LG Japan Lab Inc., LG Yokohama Innovation Center 1-2-13, Takashima, Nishi-ku, Yokohama-shi, Kanagawa 220-0011, Japan; yasukazu.yoshida@lgjlab.com

**Keywords:** soft actuator, polypyrrole, linear actuator

## Abstract

In recent decades, significant research efforts have been devoted to studying various types of actuators. Of particular interest are soft actuators based on electroactive polymers, which offer low operating voltage, light weight, and fast response. In this study, we demonstrate the feasibility of a soft linear actuator fabricated from polypyrrole (PPy), an electroactive polymer that is easy to synthesize, cost-effective, and biocompatible. By optimizing the polymerization conditions, the operation condition and environment, we were able to achieve improved and stable actuator performance. Furthermore, we developed a new actuator-contained component with a flexible counter electrode to build an actuator that operates in air. This approach enabled us to build small and lightweight actuators that operate in air, with a diameter of 5 mm, resembling artificial muscles. Our resulting miniaturized and integrated linear PPy-based actuators can be driven at low voltages (±1.5 V), making them suitable for use in various parts of the body. As such, this actuator holds significant potential for a wide range of applications in the fields of soft electronics, drug delivery, artificial organs, and muscles, as well as a component material for portable medical sensors and devices.

## 1. Introduction

An increasing aging population is a significant demographic trend in most developed countries, and it is especially pronounced in the EU and Japan. For example, the proportion of individuals aged sixty-five or older in the EU is projected to increase from 16% to 29% of the total population between 2010 and 2060 [1]. This trend will increase the social expenditure needed for the care of the aging population, placing a greater burden on the working-age population. However, this burden could be lessened if healthy older individuals and disabled people can remain independent with the aid of adequate care and support [2].

Recent technological advancements in the Internet of Things, artificial intelligence, and wearable devices have made it possible to take care of the aging population using enhanced robotics and exoskeletons [3,4,5]. However, traditional assistant robotics are heavy and noisy due to the large motors required for their operation, even though they provide high power. Soft actuators have recently been developed as an alternative to traditional assistant robotics, because they are lightweight, silent, and run on low input voltages [6,7,8,9]. Furthermore, such soft actuators are being applied more widely in microfluidic systems, biosensing systems, soft robotics, and artificial muscles [10,11,12,13]. Nonetheless, the continued development of soft actuators requires high power and strain with low driving voltage using inexpensive and lightweight materials. Additionally, it is necessary to meet the demands of commercial applications, such as high performance, device miniaturization, and long-term stability.

In recent years, there has been a significant focus on developing polymers for use as actuator materials [14,15,16,17]. In accordance with this, we have developed a soft actuator utilizing the electroactive polymer polypyrrole (PPy), which possesses facile fabrication, low cost, biocompatibility, and environmental friendliness. To obtain a durable and stainable free-standing film of PPy as an actuator, we first synthesized PPy through the electrochemical polymerization of pyrrole. The use of electrochemical deposition of PPy on an electrode allows for easy control of film thickness and morphology by adjusting polymerization conditions, such as polymerization time or solution based on electrolytes and solvents. Various studies have reported that the choice of electrolyte and solvent for polymerization significantly influences the morphology of electrochemically polymerized polypyrrole (PPy) and its actuator characteristics. The most commonly employed electrolytes in preparing polypyrrole films are tetrabutylammonium trifluoromethanesulfonate (TBACF_3_SO_3_) [18,19], lithium trifluoromethanesulfonate (LiTFSi) [20], tetra-n-butylammonium tetrafluoroborate (TBABF_4_) [21,22], and tetra-n-butylammonium bis(trifluoromethylsulfonyl)imide (TBATFSI) [23]. The polymerization solvents include dimethyl phthalate, methyl benzoate, ethyl benzoate, 1,2-dimethoxyethane, propylene carbonate, 1-Octanol, and acetonitrile [19,20,21,22,23]. These works were carried out under different conditions, such as performing temperatures and kinds of electrolytes. We chose the promising electrolytes and solvents from various previous studies and combined them to obtain six different polymerization solutions to perform optimal actuators. Additionally, we considered the effect of driving frequency and voltage on the displacement response of the actuators. The choice of counter electrode is also crucial in preventing electrolyte degradation and ensuring stable actuator operation. Therefore, we also conducted a study on the effects of electrode size on actuation performance.

Based on these results we obtained, we aimed to further our research by investigating the possibility of miniaturizing our devices, with the goal of developing an integrated actuator system that could operate in air. To achieve this goal, we conducted further experiments to electrochemically synthesize PPy on conductive fabric, which was utilized as a thin and flexible counter electrode with a thickness of 50 μm. This approach yielded small and lightweight integral actuators that have the potential to be utilized not only for artificial muscles but also as a component material for medical devices, including pacemakers, intraocular lenses, insulin pumps, and implantable medical devices [24,25].

## 2. Materials and Methods

### 2.1. Materials

For our experiments, we obtained pyrrole monomers (98%, MW 67.09 g/mol), Tetrabutylammonium tetrafluoroborate (TBABF_4_) from Sigma Aldrich (Tokyo, Japan), and Tetraethylammonium bis(trifluoromethylsulfonyl) imide (TEATFSI) from Kishida Chemical. Additionally, 2-Phenoxyethanol (PE), Ethyl Acetate (EA), and Diethyl Phthalate (DEP) were purchased from Tokyo Chemical Industry (TCI, Tokyo, Japan) as polymerization solvents. All the chemicals were used without further purification. We used TBABF_4_ and TEATFSI as electrolytes to enable the electrochemical polymerization of pyrrole. We also used PE, EA, and DEP as solvents to dissolve the TBABF_4_ and TEATFSI electrolytes and the pyrrole monomers for efficient polymerization.

### 2.2. Preparation of PPy Films

The soft actuator material used in this study was PPy film, which was obtained through constant current electropolymerization. As shown in Table 1, two types of mixed solvents were employed, a mixture of phenoxyethanol (PE) and ethyl acetate (EA) and a mixture of PE and diethyl phthalate (DEP), each mixed in a 1:1 volume ratio. To prepare the polymerization solution, a pyrrole monomer (0.15 M) and an electrolyte (0.3 M) were dissolved in a mixed solvent. The electrolytes utilized in this study consisted of three types: TBABF_4_, TEATFSI, and a mixed electrolyte in which the aforementioned two electrolytes were mixed at a molar ratio of 1:9. The morphology of PPy films has been found to have a significant correlation with the expansion ratio and generated force of the fabricated actuator. It is also well-known that the morphology of PPy films is greatly affected by the solvent and electrolyte in the electropolymerization solution [26,27,28,29,30,31]. In order to investigate the effects of these factors, six different electropolymerization solutions were prepared by combining three types of electrolytes and two types of solvents, as shown in Table 1.

To obtain PPy films with consistent morphologies, the electropolymerization temperature was maintained at −10 °C, and Ni metal plates (3 × 4 cm, thickness 0.5 mm) were utilized as the working and counter electrodes, respectively. The distance between the electrodes was fixed using plates and spacers made of PTFE (Polytetrafluoroethylene or Teflon) at a distance of 6 mm. A schematic diagram of the experimental setup is depicted in Figure 1. The polymerization time was determined based on the current density, which was set at 0.4 mA/cm^2^ in PE + EA and 0.1 mA/cm^2^ in PE + DEP. To ensure that each film had the same total amount of charge, the electropolymerization of pyrrole was carried out for 5 to 20 h. To facilitate the peeling of PPy films from the Ni electrode without causing any damage, the PPy films were soaked and swelled in acetone. The swelled PPy films could be easily peeled off from the Ni electrode, yielding free-standing PPy films that were utilized as actuator films, as shown in Figure 1.

### 2.3. Characterization of Actuator Films

The PPy films, which were electropolymerized galvanostatically as described above, were cut into pieces with a size of 10 mm width and 20 mm length for the evaluation of the actuator performance. PPy actuators were then fixed onto frames made of PTFE and suspended in an electrolyte solution, as depicted in Figure 2. The electrolyte solution consisted of lithium bis(trifluoromethanesulfonyl)imide (LiTFSI) and a mixture solvent of acetonitrile and distilled water at a volume ratio of 4:6. In ion-driven actuators, the ionic conduction is the rate-limiting factor for the reaction rate. Therefore, an aqueous solution with high ionic conduction is thought to be advantageous. However, the PPy film does not swell at all in an aqueous solution. To achieve a high expansion ratio, we aimed to swell the actuator by mixing an organic solvent such as acetonitrile. The expansion and contraction behaviors of the PPy actuator were performed using potentiostat/galvanostat (model: HA-151B, Hokuto Denko, Tokyo, Japan) and multifunction synthesizer (model: WF1946, NF Corporation, Yokohama, Japan) and measured using a laser displacement sensor (model: LK-G407, KEYENCE, Osaka, Japan) at input voltages of ±1.5, ±2.1, and ±2.4 V.

### 2.4. Preparation and Evaluation of Atmospheric-Operable Linear Actuators

To create an atmospheric-operable linear actuator, we attempted to fabricate a cylinder-like-shaped PPy film. A Ni wire with a diameter of 1.2 mm and a length of 12 cm was used as the working electrode. The polymerization solution consisted of a mixture of PE and DEP solvents with TEATFSI and TBABF_4_ as electrolytes, and it also contained 0.15 M Pyrrole. The electrochemical polymerization temperature was kept at −10 °C, as mentioned above. The current density was set at 0.1 mA/cm^2^, and the polymerization was conducted for 20 h. After the polymerization was completed, the cylindrical PPy was swollen with acetone and then removed from the Ni wire. To preserve its shape, the removed PPy tube was dried by transferring it into a plastic rod (Appendix A).

As a counter electrode, we aimed for a lightweight and easily customizable flexible electrode. Among various types of flexible electrodes, the conductive nonwoven fabric developed by 3M Inc. (model: 3M™ CN-4490, Minnesota, United States) exhibited high conductivity with a resistance of 0.005 Ω and a thickness of 50 μm. We also successfully electrochemically polymerized PPy on this electrode and used the PPy composite flexible electrode as a counter electrode. To prevent short circuits between the actuator and the counter electrode, we used a separator film developed of TORAY (model: SETELA™, thickness of 100 μm). The actuator was assembled by placing a PPy-film-based flexible electrode as the counter electrode, a separator film, and a PPy actuator inside a cylindrical container with a diameter of 5 mm.

## 3. Results

### 3.1. Effect of Polymerization Solution on Actuator Strain

To ensure high actuator performance, it is important to design a conducting polymer film with optimal properties. For PPy films, the conductivity, bulkiness, porosity, and stiffness, which are closely related to actuator performance, can be determined by polymerization conditions. Previous studies have reported that the desired actuator morphology can be obtained by controlling the polymerization conditions such as electrolyte, solvent, and temperature [32,33]. Accordingly, we chose two electrolytes and two solvents that have been previously used and proven to be promising in various studies [34,35,36] and used them in six different combinations to prepare PPy films (Table 1). The electrochemical polymerization process to obtain PPy film actuators was carried out under the same conditions as described above. The strain responses of the resulting PPy actuators prepared in different solution compositions were examined under sine-wave voltage with ±1.5, ±1.8, ±2.1 V and ±2.4 V. The results are presented in Figure 3, which shows the strain behavior of the PPy actuators under different input voltages and frequencies. As seen in Figure 3a, the strain increases with an increase in input voltage. This can be attributed to the electrochemical redox reaction mechanism that drives the PPy actuators. When the PPy backbone undergoes electrochemical oxidation, it becomes positively charged, and the anion from the electrolyte is incorporated into the film as a dopant, leading to elongation of the PPy main chain [37,38,39,40,41,42]. Accordingly, it is revealed that the polymerization conditions, especially the choice of electrolyte and solvent, have a significant impact on actuator performance. Figure 3b shows the strain response of the PPy actuators under a constant input voltage of ±2.4 V but at different frequencies. As expected, the strain decreases with an increase in frequency due to the limited response time of the PPy actuators. These results suggest that the design of a conducting polymer film with optimal properties is crucial for achieving high actuator performance. In other words, the choice of polymerization conditions, specifically the electrolyte and solvent, has a significant impact on the performance of the PPy actuator. The conductivity and morphology of the resulting polymer film are largely determined by the polymerization solution. Among the various PPy films prepared using different solution compositions, the actuator obtained from polymerization solution III-2 exhibits one of the highest levels of linear displacement reported among linear soft actuators based on conducting polymers. The SEM images of the PPy film obtained from polymerization solution III-2 are shown in Figure 4a. It can be observed that the PPy film has a highly bulky structure, which is the desired morphology for an actuator. Actuator performance was assessed by applying a square-wave voltage at various frequencies (0.05, 0.1, 0.2, and 0.5 Hz), and the displacement data were recorded automatically at a rate of three readings per second using a laser distance sensor. Figure 4c shows the actual frequency-dependent displacement amounts for an input voltage of ±2 V over the range of 0.05–0.5 Hz. As explained by the operating mechanism, the extension and contraction of the PPy film actuator are based on the injection and release of ions in the electrolyte, and the operation speed is also based on the diffusion rate of ions. In other words, as the driving frequency decreases and the driving time becomes longer, many anions in the electrolyte are injected into the film, resulting in a larger deformation rate. In terms of the relationship between frequency and strain, for example, at a frequency of 0.5 Hz, the strain was 1.46%, and the time required for maximum expansion and contraction was 1 s. Increasing this driving time to 10 s resulted in a strain rate of 14.28%, with strain increasing in direct proportion to driving time and a very high degree of strain reproducibility achieved. These results demonstrate that the electroactive polymer actuator based on PPy film has a high level of reliability regarding conductivity and structure due to the stable redox reaction of the conductive polymer and the unimpeded diffusion of ions near the electrode and in the electrolyte, enabling a highly reproducible and uniform motion of the PPy film actuator.

### 3.2. Effect of the Size of the Counter Electrode on Actuator Strain

To prevent overvoltage of the counter electrode, it is commonly practiced having a significantly larger area compared with the working electrodes. This leads to a reduction in the current density of the counter electrode. Therefore, the size of the counter electrode can have a substantial impact on the performance and stability of the actuator. We investigated the effect of electrode size on strain response by varying the ratio of the working electrode (actuator) to counter electrode areas as 1:1, 1:6, and 1:9. The displacement of the actuator under these conditions is shown in Figure 5.

Based on Figure 5, a two-fold increase in displacement was achieved when the area ratio of the counter electrode to the working electrode was increased by a factor of 9, resulting in a displacement of 1.4 mm compared with 0.7 mm. As a result, increasing the size of the counter electrode was found to have several benefits. It not only reduces the current density at the counter electrode but also leads to a significant improvement in displacement of an actuator. This improvement can be attributed to the facilitated injection of dopant ions, which are essential for the expansion and contraction of the actuator.

### 3.3. Calculating the Generated Force of an Actuator

We evaluated the generated force of a PPy film actuator. The actuator used for force measurement was a film with a length of 15 mm, a width of 2 mm, and a thickness of 100 μm. The force generated was measured using the apparatus illustrated in Figure 1, with weights of 20 g, 50 g, and 70 g. The generated force was defined as 1 MPa when a force of 10.197 kgf was applied per 1 cm^2^ of cross-sectional area. The cross-sectional area of the actuator used in this experiment was 0.002 cm^2^ (2 mm × 100 μm). Therefore, based on the cross-sectional area of the actuator used in this experiment, a load of 20 g corresponds to a generated force of 1 MPa. Using a PPy film actuator polymerized under polymerization solution III-2, as shown in Figure 6, the change in displacement was measured when the load (a) 20 g, (b) 50 g, and (c) 70 g under a driving voltage of ±1.5 V and a frequency of 0.1 Hz. As previously indicated, the generated force was 1 MPa for a load of 20 g, corresponding to 2.5 MPa for 50 g and 3.5 MPa for 70 g. In Figure 6a with a load of 20 g, the actuator exhibited a very stable and large strain response of 9.5–9.8%. However, as the load increased to 50 g and 70 g, a decrease in strain response was observed, particularly under a high load of 70 g, where creep phenomenon occurred, and the actuator did not return to its original position. There are two types of creep: mechanical creep, which occurs due to plastic deformation caused by changes in polymer conformation or molecular slippage under constant load, and electrochemical creep, which occurs due to repeated electrochemical oxidation-reduction reactions. When it comes to mechanical creep, as in this study, possible solutions include producing a harder and flatter film or increasing the film thickness through electrochemical polymerization. However, implementing such solutions unilaterally is difficult, as they may negatively impact the strain response of the actuator due to the trade-off between strain and the aforementioned solutions. Conductive polymer actuators, such as the one used in this study, may not exhibit the same level of force as metal-based actuators. Nevertheless, their film-type configuration allows for easy fabrication and molding into various shapes. By stacking and combining these film-type actuators, the force level should be increased.

### 3.4. Cycle Characteristic Based on Expansion and Contraction of an Actuator

Figure 7a presents the cyclic performance evaluation of the actuator fabricated as described earlier. The actuator, composed of a 15 mm length, 5 mm width, and 60 µm thick film, was operated continuously for one hour under a 20 g load, a driving voltage of ±1.5 V, and a frequency of 0.1 Hz. The cyclic stability of the electrochemical actuator, which denotes its repeated expansion and contraction behavior, indicates its ability to maintain stable performance over numerous cycles. Throughout the operation, the displacement of the actuator was monitored, and Figure 7b shows the displacement as a function of time and the number of cycles. In the initial stage of operation, slight creep was observed, but it was confirmed to be stabilized within a few minutes. It is considered that the observed creep phenomenon during the initial driving stage of the PPy film actuator is attributable to the change in the polymer conformation of PPy from its electrochemically polymerized structure to an optimal structure for expansion and contraction under load. For the practical application of the actuator, as indicated above, it is necessary to mitigate the initial creep phenomenon. Therefore, it is recommended to introduce a stabilization process as a preprocessing step before actual operation. Despite this aforementioned challenge, the demonstrated actuators have the capability to achieve stable expansion and contraction behavior over hundreds of cycles. This makes them promising for deployment as actuators in lightweight devices, such as flexible devices, where the use of traditional actuators using motors may be impractical.

### 3.5. Fabrication of an Atmospheric-Operable Linear Actuator

This type of ion-driven actuator requires sufficient electrolytes and a large-sized electrode surface area for ideal operation, as the driving mechanism is based on the injection and extraction of ions. However, considering various applications such as artificial muscles or flexible devices, the size of electrode and the amount of electrolyte are very limited for a practical use [43,44,45,46,47,48]. In addition, to make it adaptable for use anywhere, there is a need for actuators that can operate independently in air. Therefore, we developed an atmospheric-operable actuator device capable of efficient expansion and contraction despite the constraints of limited space. To begin with, to miniaturize and reduce the weight of the actuator, a lightweight and high-performance counter electrode is required compared with conventional metal-based counter electrodes. In recent years, various types of flexible electrodes have been commercialized with advancements in flexible devices. Herein, we attempted to synthesize PPy film by an electropolymerization on various types of flexible electrodes. However, among these, the flexible electrodes prepared by electroplating were detached and decomposed when an electric field was applied to prepare the PPy film, resulting in a loss of conductivity. Among various types of flexible electrodes, the conductive nonwoven fabric developed by 3M Inc. (3M™ CN-4490, St. Paul, MN, US) exhibited high conductivity, with a resistance of 0.005 Ω and a thickness of 50 μm, making it an ideal electrode for lightweighting and increasing flexibility of actuators. We synthesized a PPy film on this flexible electrode using electrochemical polymerization, following the same procedure as previously mentioned. In Figure 8, the structure and schematic diagram of the atmospheric-operable linear actuator are presented. The actuator was assembled by placing a PPy-film-based flexible electrode as the counter electrode, a separator film, and a PPy actuator inside a tube container with a diameter of 5 mm. A photograph of the assembled actuator is shown in Figure 8a. All of these components are thin and flexible, allowing for easy cutting into different sizes and shapes as needed. The actuator was fabricated by electrochemically polymerizing PPy onto a Ni wire, as illustrated in Figure 8a. The electrochemically polymerized PPy film on the Ni wire was soaked and swelled in acetone, and then the Ni wire was removed. The resulting cylindrical free-standing PPy structure was utilized as the actuator. To prevent short circuits between the actuator and the counter electrode, a separator film, commonly used in batteries was inserted as an insulating sheet. The separator film allows for the charge transfer while effectively preventing short circuits, thereby ensuring the smooth operation of the actuator. Figure 8b illustrates the relationship between the frequency of actuation and strain, as well as the peak-to-peak time and strain. The relationship between actuation time and strain was found to be linearly proportional, with strains of 0.29% and 2.95% observed at actuation times of 1 s and 10 s, respectively. However, the cylindrical actuator exhibited lower strains compared with the film-type actuators. The cylindrical shape of the actuator may be one of the reasons for the lower strain compared with film-type actuators, despite being fabricated under the same electropolymerization conditions. It is believed that the cylindrical shape of the actuator leads to uneven distribution of dopant ion injection between the inner and outer parts of the cylinder, resulting in nonuniformity, such as the presence of low-elongation areas in the actuator and a decrease in the overall strain rate. Although there was a decrease in strain, we successfully obtained a lightweight and stable linear actuator capable of operating in air by modifying the structure and composition of the actuator.

## 4. Discussion

In this study, by optimizing the polymerization conditions of the actuator polymer, polypyrrole, we successfully obtained a highly stretchable soft actuator. The polymerization conditions have a significant impact on the morphology, conductivity, and molecular structure of the actuator polymer, but obtaining the optimal conditions was challenging due to the trade-off relationship among the actuator properties. Furthermore, by designing the counter electrode and the structure, we achieved the fabrication of an atmospheric-operable actuator. Although the stretchability is not yet sufficient, we believe that further improvements can be achieved by refining factors such as electrode arrangement to improve both stretchability and stability.

Compared with conventional actuators powered by small motors, hydraulics, or pneumatics that are commonly used in industrial robots and machinery, soft actuators based on polymer materials provide significant advantages of being compact, lightweight, and operating silently. Electroactive polymer actuators, which respond to electrical stimuli, have demonstrated the ability to operate at low voltages and achieve reliable and controllable deformation. Although the deformation of the actuator demonstrated in this study is still relatively small compared with motor-driven actuators, the miniaturization, lightweight design, and atmospheric-operability achieved with only a 5 mm diameter demonstrate the potential for application in biomimetic muscles and other biologically inspired actuators. The ability to operate at low voltages without motor noise or heat generation further enhances its suitability for such applications. Furthermore, the high biocompatibility of actuators using polypyrrole opens up possibilities for their application in various fields, including artificial muscles, as well as catheters, guide wires, patch insulin pumps, and autofocus-enabled contact lenses.

## Figures and Tables

**Figure 1 polymers-15-03455-f001:**
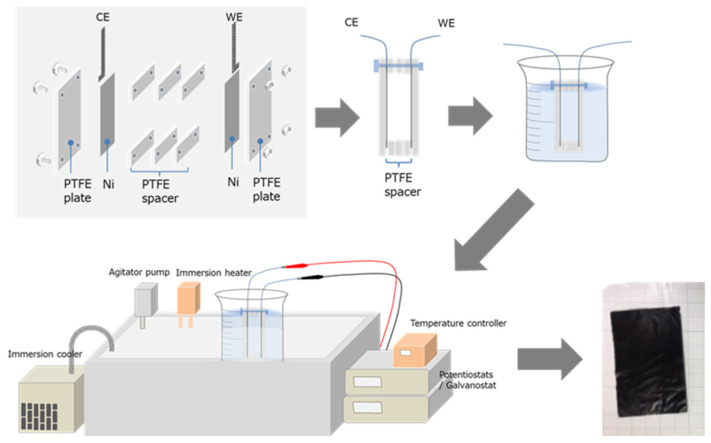
The experimental setup used for the preparation of PPy actuator film: Ni metal plates (3 × 4 cm, thickness 0.5 mm) were used as the working and counter electrodes, and the distance between the electrodes was fixed with PTFE plates and spacers at a distance of 6 mm. Six different electropolymerization solutions were used to obtain PPy films, and the polymerization solution was controlled at −10 °C during polymerization. The photograph of the obtained free-standing PPy films is presented.

**Figure 2 polymers-15-03455-f002:**
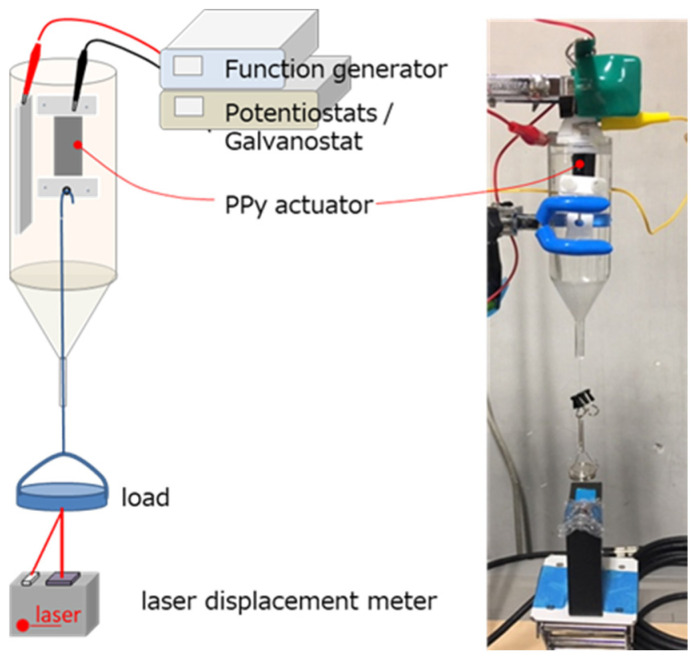
(**Left**) Schematic illustration of the experimental setup used for measuring the performance of the linear actuator based on PPy films. (**Right**) the photograph of experimental setup during the expansion/contraction measurement of PPy actuator. The size of the actuator film is 10 mm in width and 20 mm in length. The stretching characteristics were evaluated using mixed solvent of acetonitrile and distilled water with lithium bis(trifluoromethanesulfonyl)imide (LiTFSI) as the electrolyte.

**Figure 3 polymers-15-03455-f003:**
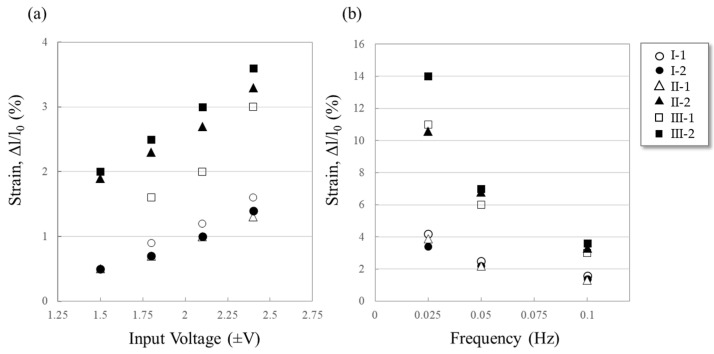
Comparison of strain (%) among PPy film actuators prepared using 6 different polymerization solutions (**a**) at different input voltage and (**b**) different frequency at ±2.4 V. The strain was calculated from dl/l_0_ (dl = change in length, l_0_ = initial length).

**Figure 4 polymers-15-03455-f004:**
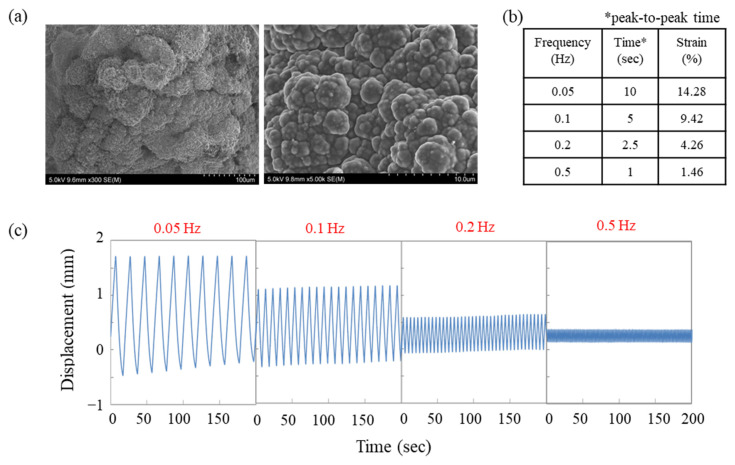
(**a**) SEM images of PPy film polymerized under polymerization solution, III-2 in Table 1, and (**b**) the strain and (**c**) displacement response of linear actuator at square voltage of ±2 V and frequencies from 0.05 to 0.5 Hz. The displacement was monitored by CCD laser displacement sensor.

**Figure 5 polymers-15-03455-f005:**
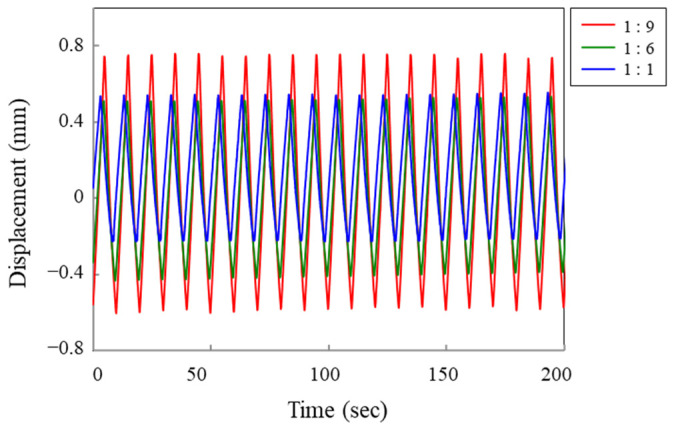
Linear displacement response of an actuator as a function of the size ratio between the working and counter electrodes, with ratios of 1:9, 1:6, and 1:1, respectively.

**Figure 6 polymers-15-03455-f006:**
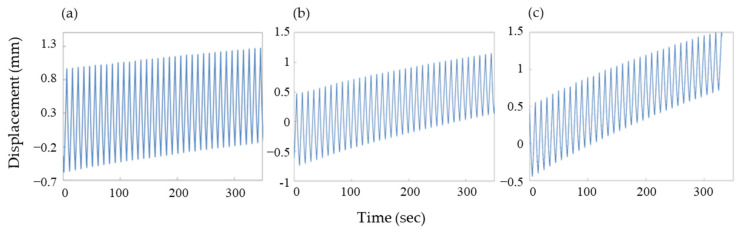
Changes in displacement for linear PPy actuator under load of (**a**) 20 g; (**b**) 50 g; and (**c**) 70 g with driving voltage of ±1.5 V and frequency of 0.1 Hz.

**Figure 7 polymers-15-03455-f007:**
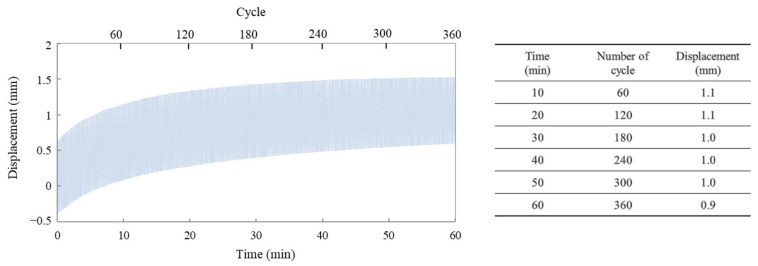
Cycle life of the linear actuator under continuous operation at ±1.5 V and 0.1 Hz.

**Figure 8 polymers-15-03455-f008:**
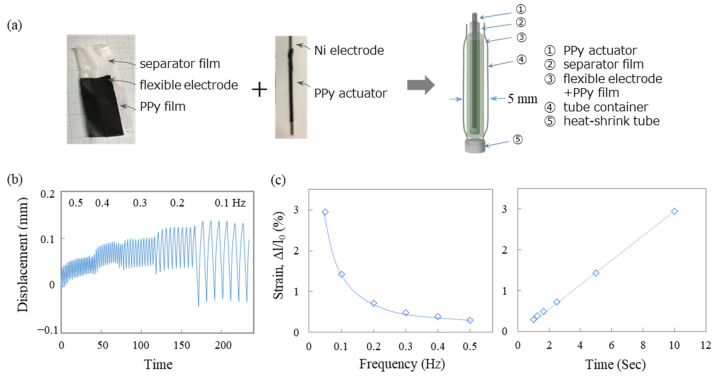
(**a**) Photograph and schematic illustration depicting the structure and assembly procedure of the actuator; (**b**) displacement versus time curves; (**c**) strain versus frequency and peak-to-peak time of the integrated actuator.

**Table 1 polymers-15-03455-t001:** The polymerization solutions used in this study were of six diverse types, each formed by combining three types of electrolytes with two types of solvent. By varying the types of electrolytes and solvent used, six distinct polymerization solutions were obtained and utilized for the synthesis of PPy film.

Electrolyte	Solvent	(Abbreviation)
TBABF_4_	PE + EA	I-1
	PE + DEP	I-2
TEATFSI	PE + EA	II-1
	PE + DEP	II-2
TBABF_4_ + TEATFSI	PE + EA	III-1
	PE + DEP	III-2

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
