# Peer review of "A Lightweight and Low-Voltage-Operating Linear Actuator Based on the Electroactive Polymer Polypyrrole"

_polymers, 2023, doi:10.3390/polym15163455_

Round 1

Reviewer 1 Report

This manuscript reports the polypyrrole-based actuator. Polypyrrole was electropolymerized under the conditions of six combinations of electrolyte and solvent. The authors picked one best polymerization condition and tested actuator performance both in electrolyte solution and in air. For the publication, following concerns need to be addressed

 1. The authors used 6 PPy polymerization condition using two electrolytes and solvent mixture. Is these conditions are by the authors for the first time? Please compare the conditions used in this study with the other polymerization conditions used in other literatures and describe the difference and originality of this study.

2. The authors said that condition III-2 showed best results owing to highly bulky structure, as shown in SEM. Please give the SEM images of other 5 polymerization conditions. Also, explain why do you think III-2 shows best results. Did you also measured the conductivity of 6 samples?

3. Figure. 8. There is no legend for Figure 3c

4. Please describe the fabrication and evaluation procedure of atmospheric-operable linear actuator in experimental section, including the dimensions and model names of materials used, diameter of Ni electrode, separator film model name and thickness, etc.

5. Why do you need heat-shrink tube in Figure 8a. How do you measure the length change the actuator of Fig. 8a.

6. The authors mentioned, driving mechanism of ion-driven actuator, injection and extraction of ion. Please be kind to explain the driving mechanism of atmospheric-operable actuator, too. You mentioned dopant ion injection in line 285 and 286. Is this related to the driving mechanism of atmospheric-operable actuator?

7. line 273, how did you removed Ni wire? Did you rolled the ppy film

8. Please give information about the equipments such as potentiostat and laser displacement meter.

Author Response

Dear Reviewer,

Thank you very much for your thoughtful review. Based on your comments, we have carefully gone over the manuscript and incorporated the suggested changes. We have uploaded the final version of the manuscript, highlighting some parts with changes made in red characters. We hope the revised manuscript meets your expectations and requirements. We truly appreciate your valuable feedback. Thank you once again.

Reviewer 2 Report

1. Polypyrrole has been widely used to fabricate actuators in all kinds of configurations. The materials, fabrication method and actuator design in this manuscript have been reported previously. The authors should highlight in the introduction part the novelty or the new insight provided in the manuscript. 

2. Is there any reason to use TBABF4/TEATFSI mixed electrolyte with a molar ratio of 9:1? What happens if you change the molar composition?

3. Figure 4a shows the PPy morphology made from III-2. What does the morphology of PPy look like if you use other polymerization solutions?

4. Can you explain the origin of the large baseline shift in the displacement measurement? 

5. Can you envision some of the application scenarios of the PPy-based electromechanical actuators, considering their slow strain response and significantly frequency-dependent strain level? 

Author Response

(The authors gave the same response as above.)

Round 2

Reviewer 1 Report

The authors successfully revised the manuscripts according to the comments.

There is one point that is unclear about comment #6.  

I am asking if there is difference in mechanism between ion-driven actuator  which require electrolyte and atmospheric-operable actuator. 

Author Response

There is one point that is unclear about comment #6. 

I am asking if there is difference in mechanism between ion-driven actuator  which require electrolyte and atmospheric-operable actuator.

Response: We appreciate your feedback and apologize for any insufficient explanation in our previous response regarding the paper review. There is no difference in the operating mechanism between an atmosphere-operable actuator and other ion-driven actuators. To create atmosphere-operable actuators that are lightweight and compact, similar to muscles, a cylindrical-shaped actuator is used, along with lightweight flexible electrodes employed as counter electrodes as previously mentioned.

Reviewer 2 Report

I have no further questions and would recommend the publication of this manuscript. 

Author Response

I have no further questions and would recommend the publication of this manuscript. 

Thank you very much for your thoughtful review and feedback on our manuscript. We truly appreciate your time and expertise in evaluating our work.